# Research on the Structure of Carbon Emission Efficiency and Influencing Factors in the Yangtze River Delta Urban Agglomeration

**Chenxu Liu** [1,*], **Ruien Tang** [1], **Yaqi Guo** [1], **Yuhan Sun** [1] **and Xinyi Liu** [2]

[1] School of Geography Sciences, Nanjing Normal University, Nanjing 210023, China; 26180409@njnu.edu.cn (R.T.); 10180240@njnu.edu.cn (Y.G.); 10200122@njnu.edu.cn (Y.S.)
[2] Honorary College, Nanjing Normal University, Nanjing 210023, China; 01210230@njnu.edu.cn
[*] Correspondence: liuchenxu0216@126.com

**Abstract:** Climate change caused by $CO_2$ emissions has become one of the most serious environmental problems facing the world today, and it has a strong relevance to sustainability. This paper measures the carbon emission efficiency of the Yangtze River Delta urban agglomeration from 2001 to 2019 using the U-S SBM model. The modified gravity model and social network analysis methods are used to explore its spatially correlated network structure, and QAP regression is used to explore the influencing factors. The results show the following: (1) The spatial correlation of the carbon emission efficiency in the Yangtze River Delta urban agglomeration increased during the study period, showing a complex network structure with multiple threads and directions, and a strong mobility of the network. (2) The spatial network of the carbon emission efficiency in the Yangtze River Delta urban agglomeration gradually formed a core−edge structure with southern Jiangsu as the core area, northern Zhejiang and central Jiangsu as the secondary core area, and central Anhui and southern Zhejiang as the edge area during the study period. (3) The spatial correlation network of carbon emission efficiency in the Yangtze River Delta urban agglomeration is divided into "net benefit", "net spillover", "two-way spillover", and "broker". (4) Differences in energy intensity, government environmental regulations, technology research and development, and economic export orientation are the main factors affecting the spatial correlation of carbon emission efficiency in the Yangtze River Delta urban agglomeration.

**Keywords:** carbon emission efficiency; spatial connection; social network analysis; influencing factors

## 1. Introduction

Climate change and its impacts have become one of the most serious environmental problems facing the world today [1]. In its fourth Global Climate Assessment, the United Nations Intergovernmental Panel on Climate Change (IPCC) noted that it is an indisputable fact that human activities and massive greenhouse gas emissions are the major causes of global climate change. As $CO_2$ is one of the most important greenhouse gases, it is closely related to global warming [2]. As the main source of carbon emissions, cities have a profound impact on the realization of carbon emission reduction targets [3]. Establishing low-carbon cities is an inevitable choice for China in order to deal with climate change and to develop a low-carbon economy [4].

Based on the above, carbon emission and carbon emission reduction issues have been given extensive attention, and studies on carbon emission estimation methods [5], influencing factors [6–8], emission intensity [9,10], and emission efficiency [11] have been carried out successively. Carbon emission efficiency is an important concept in environmental science; it refers to the economic benefits generated by production activities that produce carbon emissions at the same time [12]. The less carbon emissions generated per unit of economic output, the more carbon emission efficient it is. Carbon emission efficiency

considers the promotion and inhibitory effect of carbon emissions on economic growth, and measures the level of economic growth under carbon emission restrictions—this has been widely studied. In the context of tightening carbon dioxide emission constraints, the currently used crude economic development approach is unsustainable, and improving carbon efficiency is an important way to promote a change to the development approach. At present, most studies focus on regional differences in carbon emission efficiency, and less attention has been paid to region-specific carbon emission efficiency correlations.

As a typical representative of the world's major urban agglomerations and metropolitan areas, from an early stage, the Yangtze River Delta region advanced industrialization and urbanization, and is home to a large number of high-emission industries such as petrochemicals, metallurgy, paper making, and automobiles. The energy and carbon emissions brought about by the high-intensity development of industry have put enormous environmental pressure on the Yangtze River Delta urban agglomerations (Figure 1), affecting the sustainable development of the region and the fulfilment of emission reduction commitments.

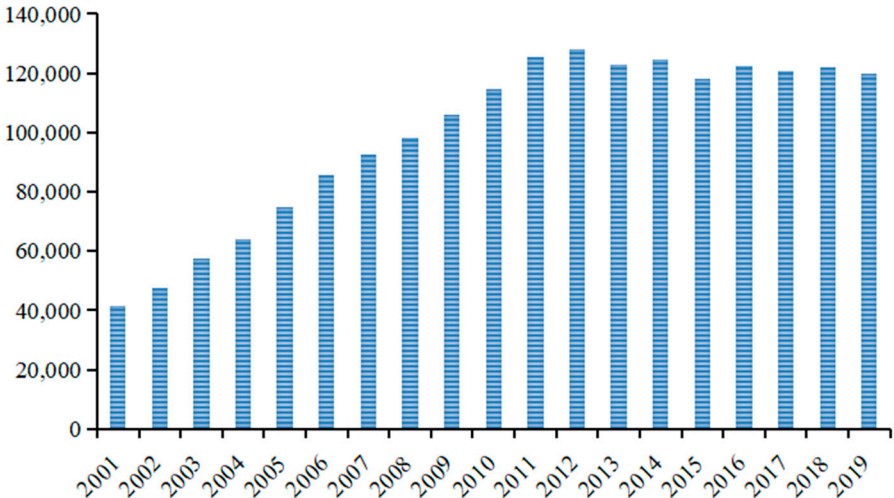

**Figure 1.** Carbon emissions in the Yangtze River Delta urban agglomeration from 2001 to 2019 (source: extracted from the DMSP/OLS night light data).

At the same time, the gradual formation of a networked transportation system, the continuous development of information technology, and the continuous advancement of regional economic integration have made the spatial connection between the production factors of each city in the Yangtze River Delta increasingly close, and the carbon emission efficiency among the regions has also shown significant spatial correlation characteristics. It is of great theoretical significance and application value to examine the spatial correlation structure and influencing factors of energy carbon emission efficiency in the cities of the Yangtze River Delta urban agglomeration from a network perspective, and to explore the status and role of carbon emission efficiency of each region in the spatial correlation network, in order to build a cross-regional carbon emission efficiency synergy mechanism under the new economic normal and to formulate carbon emission reduction policies that are both targeted and systematic. It can also fill the gap in the current academic field in the study of the spatial correlation of regional carbon emission efficiency.

Based on the background, this paper aims to address the following three questions: (1) Explore the spatial association of carbon emission efficiency in the Yangtze River Delta urban agglomeration and describe the structural characteristics of this spatial association using social network analysis and other econometric methods. (2) Explore the factors influencing the spatial association of carbon emission efficiency in the Yangtze River Delta urban agglomeration and analyze the influence mechanisms of each variable. (3) Propose

policy implications based on the research results and discuss the practical application value of building a synergistic carbon emission reduction mechanism.

## 2. Literature Review

At present, the definition of carbon emission efficiency can be divided into two types: single factor carbon emission efficiency and total factor carbon emission efficiency. Kaya and Yokobori first defined carbon emission efficiency as carbon productivity from a single-factor perspective; that is, the ratio of GDP to carbon emissions in the same period [13]. Yamaji defined the ratio of total $CO_2$ emissions to GDP as $CO_2$ productivity when studying the level of carbon emissions in Japan [14]. Mielnik, Goldember, and Ang used carbon dioxide emissions per unit of energy consumption as an important measure of carbon emission efficiency [15,16]. The single factor only takes into account the proportion between GDP or energy consumption and carbon emissions, but ignores the substitution between factors when multiple factors combined are input into the actual production process. Ramanatha believed that the definition of carbon emission efficiency should be integrated into the three frameworks of energy consumption, economic development, and carbon emission, so that the evaluation results are comprehensive and reasonable [17]. Zaim and Taskin defined carbon emissions as a non-expected output variable, and proposed the concept of the comprehensive efficiency index, and applied this index to the OECD national research [18].

The current research on carbon emission efficiency can be divided into industries and regions according to the research objects. In the research of industry carbon emission efficiency, scholars have used different measurement models to measure the carbon emission efficiency of different industries in the national economy. Wang Kai and Wang Kun used the SBM model, and found that the carbon emission efficiency of China's tourism showed a significant spatial imbalance [19]; Dwyer et al. measured carbon emissions from tourism in Australia using both the production and expenditure approaches [20]; Hampf proposed a new DEA analysis method based on an efficiency analysis perspective to investigate the standard of $CO_2$ emissions in the U.S. electric power industry [21]; and Erwin et al. used a sample of Indonesian manufacturing firms to explore the determinants of carbon emissions [22]. In terms of regional carbon emission efficiency research, Ramanathan used the data envelope analysis (DEA) to build an input−output index system containing carbon dioxide emissions, energy consumption, and economic activity variables, to compare the carbon emission performance level of various countries [17]. Zhang et al. developed a spatial regression model to study the convergence characteristics and influencing factors of carbon emission intensity in Chinese cities and major strategic regions [23]. Meng et al. used the RAM-DEA model to estimate the low-carbon economic efficiency of the Chinese industrial sector from 2001 to 2013, and found that most industries of low-carbon economic efficiency are still at a low level; however, the carbon emission efficiency was greatly improved during the study period [24].

In addition, many academic studies have confirmed that carbon-emission-related problems do not exist independently among regions, but they have some spatial correlations between them [25,26]. Grunewald and others explored the driving factors of spatial differences in carbon emissions and pointed out that energy intensity and energy structure are the main reasons for the spatial differences in carbon emissions [27]. Marbuah and Mensah performed a statistical test of the spatial association of several pollutants, including $CO_2$, using 290 Swedish urban areas as the study areas, showing that spatial spillover effects were the main driver of the environmental Kuznets curve [28]. Wu studied the spatial pattern and evolution mechanism of carbon emission reduction in China through spatial econometrics, and analyzed the emission reduction characteristics of key provinces [29]. Zhou determined the determinants and spatial relationship of $CO_2$ emissions at an urban level in China [30].

Previous studies have also discussed and analyzed the influence mechanism of regional carbon emission efficiency in depth. Wang et al. used the window SBM analysis method to measure the carbon emission efficiency and emission reduction potential of

various provinces and cities in China from 2003 to 2016, and analyzed the impact of resource endowment on emission efficiency using the panel Tobit model. The results show an inverse relationship between resource endowment and emission efficiency [31]. Liu et al. proposed the ideal point cross efficiency (IPCE) model, and used this model to analyze the carbon emission efficiency of the top ten urban agglomerations in China in 2008–2015. The results showed that the population effect and economic effect promoted the emission efficiency of mature urban agglomerations, while reducing the efficiency of emerging urban agglomerations [32]. Zhou et al. measured the carbon efficiency of the top 18 global carbon emitting countries from 1997 to 2004 based on a DEA model, and found that technological progress had a significant effect on the improvement of carbon efficiency [33]. Ma Y. and Lu Y. used the ultra-efficiency SBM model to calculate carbon emission efficiency at a provincial level in China from 1995 to 2012, and examined the impact of independent innovation, FDI, and international trade. The results found that FDI could significantly improve the carbon emission efficiency, while independent innovation and import had inhibitory effects [34].

Based on the above research, it can be seen that scholars at home and abroad have carried out a lot of in-depth research on the industry and regional carbon emission efficiency and carbon emission space correlation, and the existing research has achieved fruitful results, in both the research perspectives and research methods. The ultimate purpose of carbon-emission-related research is to establish an effective carbon emission reduction mechanism, which lays a solid research basis and theoretical foundation for the following research in this paper. However, on the basis of the existing studies, there are still some issues that deserve further discussion:

(1) Most of the current studies on carbon emission efficiency are conducted at national, regional, provincial, and municipal levels and at the industry level, among others. Although these studies have strong theoretical significance and practical value, it is more relevant to study the issue of carbon emissions from a specific economic and social context, taking into account regional characteristics.

(2) The spatial correlation and heterogeneity of carbon emissions have been explored and revealed to a certain extent, laying a foundation for the study of regional synergy in reducing emissions. However, most of the existing literature is based on an empirical analysis of the spatial econometric models, which only empirically examine the attribute data of carbon emissions, and do not reveal the inter-regional linkage structure from the perspective of spatial correlation, while the correlation of the carbon emission efficiency between regions is rarely addressed.

(3) Although the current studies on the influencing factors of carbon emission efficiency are attributed to the economic level, population size, industrial structure, energy structure, and technological progress, etc., there is no unified understanding of the specific driving effects of these influencing factors, and the influencing factors of the formation of the spatial network structure of carbon emission efficiency also need to be further studied. Only by clarifying the specific effects of these influencing factors on different regions from the perspective of spatial correlation can we correctly grasp the causes of regional differences in carbon emission efficiency and reveal the essential characteristics of regional carbon emission efficiency changes.

Synthesizing the above content, this paper uses the U-S SBM model for the calculation of the carbon efficiency of the Yangtze River Delta urban agglomeration from 2001 to 2019 from the perspective of spatial correlation, uses social network analysis to depict the Yangtze River Delta urban agglomeration's carbon efficiency network structure, and then provides the basis for the cross-regional carbon emission reduction policy based on these results.

## 3. Study Area

The Yangtze River Delta urban agglomeration, located in the Yangtze River estuary alluvial plain (Figure 2), according to the State Council approved of the Yangtze River

Delta urban agglomeration development plan, includes the following: Shanghai, Nanjing, Wuxi, Changzhou, Suzhou, Nantong, Yancheng, Yangzhou, Zhenjiang, Taizhou, Hangzhou, Ningbo, Jiaxing, Shaoxing, Jinhua, Zhoushan, Taizhou, Hefei, Wuhu, Maanshan, Tongling, Anqing, Chuzhou, Chizhou, and Xuancheng, which totals 26 cities. The Yangtze River Delta urban agglomeration accounts for 2.3% of the country's total area, and has a population of 225 million, contributing to about a quarter of the country's GDP.

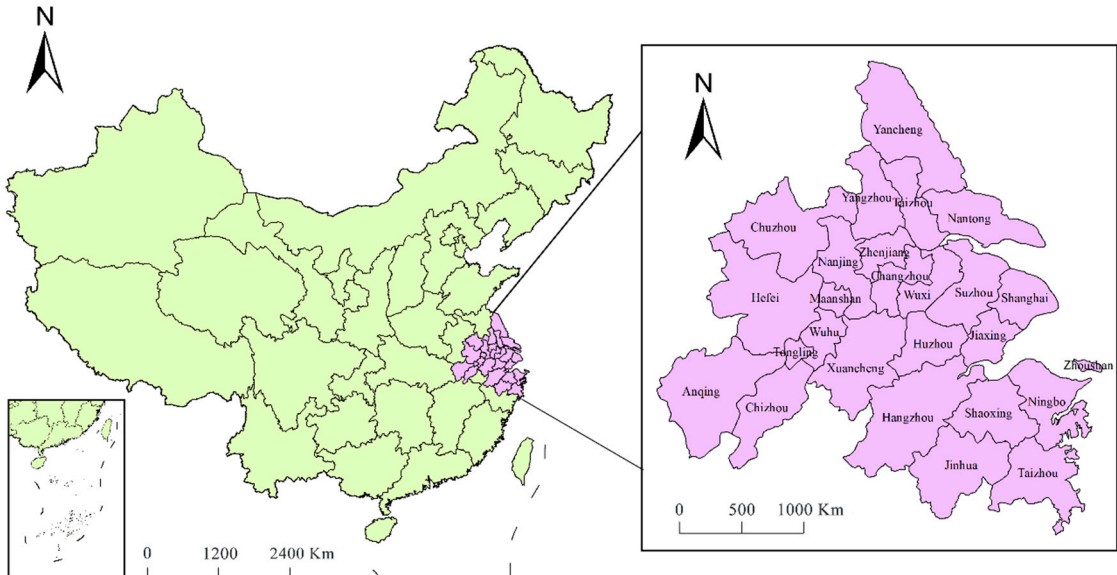

**Figure 2.** Location of the Yangtze River Delta urban agglomeration (source: authors' own work).

The industrialization level of the Yangtze River Delta region is more developed in the whole country. The rapid industrialization leads to a significant increase in its demand for energy, resulting in a large amount of carbon emissions. The contradiction between environment and economic growth in the region is very serious [26]. With the formation of the transportation network and the flow of economic factors, the urban spatial connection in the Yangtze River Delta region is constantly strengthened, and the network characteristics are becoming more and more obvious, which has the realistic necessity and structural basis for cross-regional coordinated emission reduction.

## 4. Methods

### 4.1. Calculation of Carbon Emission Efficiency

In this paper, we used the U-S SBM model to calculate the energy carbon emission efficiency of 26 cities in the Yangtze River Delta from 2001 to 2019. In the process of economic production, the input of labor, capital, and energy would not only produce industrial products, but also by-products that cause environmental pollution, which is an undesired output [35]. The U-S SBM model was first proposed by Tone [36]. Compared with the traditional data envelope model (DEA), the ultra-efficiency SBM model based on unexpected output can, on the one hand, solve the problem of input−output relaxation, and, on the other hand, solve the efficiency analysis of unexpected output. The mathematical expression form of the model is as follows:

$$\rho* = \min \frac{\frac{1}{m}\sum_{i=1}^{m}\frac{\overline{x}_i}{x_{i0}}}{\frac{1}{S_1+S_2}\left(\sum_{r=1}^{S_1}\frac{y_r^g}{y_{r0}^g}+\sum_{r=1}^{S_2}\frac{y_r^b}{y_{r0}^b}\right)}, \text{s.t.}\begin{cases}\overline{x} \geq \sum_{j=1,\neq k}^{n}\theta_j x_j\\[2mm]\overline{y^g} \geq \sum_{j=1,\neq k}^{n}\theta_j y_j^g\\[2mm]\overline{y^b} \geq \sum_{j=1,\neq k}^{n}\theta_j y_j^b\\[2mm]\overline{x} \geq x_0, \overline{y^g} \leq y_0^g, \overline{y^b} \geq y_0^b, \overline{y^g} \geq 0, \theta \geq 0\end{cases} \tag{1}$$

In Formula (1), $\rho*$ is the efficiency value of the decision-making unit; $x$, $y^g$, and $y^b$ are the input−output vectors, which constitute the evaluation decision making unit; and $S = (S\text{-}, S_g, \text{and } S_b)$ represents the relaxation of input, expected output, and unexpected output, respectively.

Combined with the characteristics of the study area, the capital, energy, and labor indexes were selected as the input factors; GDP was selected as expected output in the process of economic activities; and urban $CO_2$ emissions were regarded as the unexpected output of economic activities. The input−output index system of carbon emission efficiency of urban agglomeration in the Yangtze River Delta matching the model was established (Table 1).

**Table 1.** Input−output index system of urban carbon emission efficiency in the Yangtze River Delta.

| Index | Variable | Unit |
|---|---|---|
| Investment index | Investment in the fixed assets | CNY 100 million |
| | Employee | Thousands of people |
| | Electricity consumption | One hundred million kilowatt-hours |
| Expect output | GDP | CNY 100 million |
| Undesired output | Energy carbon emissions | Ten thousand tons |

### 4.2. Modified Gravity Model

Building the spatial correlation matrix of the carbon emission efficiency between cities and determining the relationship between regions is the premise of depicting the network structure. This paper incorporates economic and geographical factors into the framework of spatial correlation, and uses an improved gravity model to construct the spatial correlation matrix of the carbon emission efficiency in the Yangtze River Delta urban agglomeration. The expression form is as follows:

$$G_{ij} = \frac{C_i}{C_i + C_j} \times \frac{C_i \times C_j}{\frac{D_{ij}^2}{(e_i - e_j)^2}} \tag{2}$$

In Formula (2), $i$ and $j$ represent city $i$ and city $j$ in the urban agglomeration, respectively. $G_{ij}$ is the correlation degree of the carbon emission efficiency between cities. $D_{ij}$ represents the geographical distance between cities. $C_i$ and $C_j$ represent the carbon emission efficiency of city $i$ and city $j$, respectively. $E_i$ and $E_j$ represent the per capita GDP of city $i$ and city $j$, respectively.

According to the above formula, the spatial correlation matrix of urban carbon emission efficiency is calculated, with the mean of each row in the matrix as the critical value. If the association strength is greater than the mean, it is recorded as 1, which indicates a correlation between two cities; otherwise it is 0, indicating no correlation, thus forming a spatial binary matrix as the basis of the network structure analysis.

### 4.3. Social Network Analysis Methods (SNA)

According to the spatial correlation matrix of carbon emission efficiency in the Yangtze River Delta urban agglomeration calculated based on the gravity model, the structural

characteristics of the carbon emission efficiency network in the urban agglomeration were measured using the social network analysis method (SNA), which mainly included the following three parts:

(1) Overall network structure features. The overall network structure features were measured by three indicators: density, correlation degree, and efficiency of carbon emission cyberspace. Among them, the network density reflects the tightness of the whole network structure. The greater the network density, the closer the carbon emission efficiency relationship between cities. Network correlation degree reflects the connectivity between cities in a spatial network. The fewer isolated cities in the network, the greater the correlation degree of the network. Network efficiency reflects the connection efficiency between cities in the spatial correlation network. The more associations that exist between cities, the more stable the network, and the lower the network efficiency [37].

(2) Node network structure features. The structure features of the node network were measured by three indexes: degree centrality, proximity centrality, and intermediary centrality. Among them, the degree centrality represents the center degree of cities in the spatial network. Intermediary centrality reflects the ability of one city to correlate with other cities. The greater the intermediary centrality, the greater the role of "intermediaries" in the network, and thus the more significant the relevance in controlling carbon emissions from other cities. The proximity centrality depicts the degree to which a city is "not controlled by other cities". If a city has a high proximity centrality, it shows that the connection between the carbon emission efficiency of the city and other cities is mainly directly related and less controlled by other cities, and the city plays the transport function in the network.

(3) Spatial clustering features. The block model is a method used to analyze the internal structure state and the position and role of each region in the plate in the social network analysis [38]. This method divides the nodes in the network into different modules by clustering, and then analyzes the interaction of each module in the spatially related network. Drawing on relevant research [39], this paper divided the spatial correlation network of carbon emission efficiency in the Yangtze River Delta urban agglomeration into four modules, "net benefit plate", "net overflow plate", "two-way spillover plate", and "broker plate", to discuss the spatial relationship of regional carbon emission efficiency.

The above analysis procedure was implemented in Ucinet 6.0 software (This software was developed by University of California, Irvine, which located in Irvine, Southern California, it's the most popular social network analysis software for users worldwide).

*4.4. QAP Regression Method*

This study used QAP regression to analyze the influencing factors of carbon emission efficiency in the Yangtze River Delta urban agglomeration. QAP regression targets the variables with a high correlation, determines the correlation coefficient between the matrices by comparing the differences between two matrices, and finally performs a non-parametric test of the correlation coefficient. This approach can circumvent the problem of collinearity among variables in the traditional multiple regression method, thus improving the confidence of the conclusions.

Combining the relevant research [40–43] and characteristics of the research area, assuming that the energy structure (E), government environmental regulation (G), energy intensity (F), industrial structure (P), technical level (T), carbon sink efficiency (R), and external connection (O) are the factors that may affect the spatial correlation of carbon emission of the urban agglomeration, the following model is established:

$$\text{Net} = f(E, G, F, P, T, R, O) \tag{3}$$

In Formula (3), Net represents the network relationship of carbon emission efficiency in the urban agglomeration, and E, G, F, P, T, R, and O each represent the differences in various elements between cities. We took the average value of the corresponding index of each city in the urban agglomeration during the research period, and used the difference between cities to establish the difference matrix for the QAP correlation and regression analysis. The definition of each variable is shown in Table 2.

**Table 2.** Variable definitions in the QAP methods.

| Variable | Meaning | Index |
| --- | --- | --- |
| E | Differences in energy resource structure | The difference in coal consumption in total energy consumption (%) |
| G | Differences in environmental regulation | The difference in proportion of inter-urban government fiscal expenditure in GDP (%) |
| F | Differences in energy intensity | The difference between per unit of GDP (ton of standard coal/CNY 10,000) |
| P | Differences in industrial structure | The difference between the output value of the secondary industry between cities (%) |
| T | Differences in technical level | The difference between R&D expenditure (CNY 10,000) |
| R | Differences in carbon sink efficiency | The difference between per capita green area (ten thousand people/m$^2$) |
| O | Differences in external contact | The difference between total import and export trade between cities (CNY 100 million) |

*4.5. Data Source*

The carbon emission data of the Yangtze River Delta urban agglomeration involved in this study were extracted by DMSP/OLS night light data [44]. After comparison with cities with energy consumption data, the error was less than 6.7%, with good simulation accuracy. Other social and economic data were from the Statistical Yearbook, missing data were obtained by interpolation, urban distance data were calculated by distance function of ArcGIS 10.8 software, and vector base map data were from the Standard Map of Ministry of Natural Resources (GS (2019) 1825.

**5. Results and Discussion**

*5.1. Spatiotemporal Evolution Process of Carbon Emission Efficiency in the Yangtze River Delta Urban Agglomeration*

5.1.1. Analysis of Measurement Results

The energy carbon emission efficiency of the Yangtze River Delta urban agglomeration from 2001 to 2019, based on the SBM model considering non-desired outputs, was calculated as shown in Appendix A.

The results showed that, from 2001 to 2019, the energy carbon emission efficiency of the Yangtze River Delta urban agglomeration showed a fluctuating decline (Figure 3), with the average value changing from 0.671 in 2001 to 0.522 in 2019. In 2001–2004, because of the implementation of strict energy conservation and emission reduction and environmental protection initiatives in China, the carbon emission efficiency of cities was generally high, and showed an upward trend from 2005 to 2010. After 2010, with the arrival of an energy

conservation and emission reduction bottleneck, the rapid industrialization of the Yangtze River Delta region, and the impact of the economic crisis from 2004 to 2008 [45], the carbon emission efficiency decreased. After 2015, cities in the Yangtze River Delta region entered the industrial restructuring and transformation period one after another, and the carbon emission efficiency stabilized. With the implementation of the overall strategy of high-quality development and ecological civilization construction in the Yangtze River Delta urban agglomeration, the energy carbon emission efficiency of some cities increased. With the implementation of the Yangtze River Economic Belt's high-quality development and the overall strategy of ecological civilization construction, the energy carbon emission efficiency of some cities improved significantly.

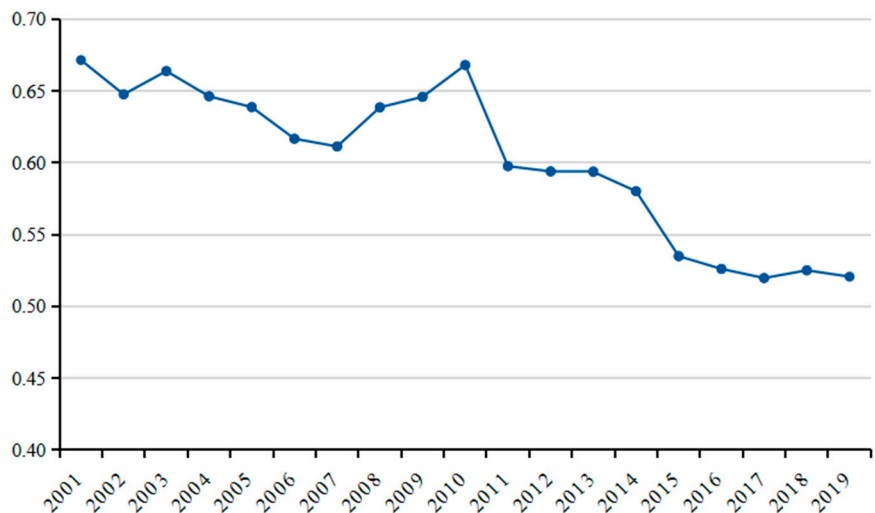

**Figure 3.** Average carbon emission efficiency of the urban agglomeration in the Yangtze River Delta from 2001 to 2019 (source: calculated from Formula (1)).

5.1.2. Spatial and Temporal Evolution Process

As shown in Figure 4, the spatial distribution of carbon emission efficiency in the Yangtze River Delta urban agglomeration gradually stabilized from 2001 to 2019, and showed significant differentiation characteristics, showing the overall distribution pattern of high in the east and low in the west.

In 2001, cities with a high carbon emission efficiency were mainly distributed in southern Jiangsu, eastern Zhejiang, and along the rivers of Anhui, while cities with a low-carbon emission efficiency were mainly distributed in eastern Anhui and northern Jiangsu. In 2010, the high carbon emission efficiency tended to expand in the north and south directions, and the carbon emission efficiency of cities within the boundaries of Yancheng, Nanjing, and Taizhou was significantly improved, and the high-efficiency cores of southern Jiangsu and northern Zhejiang were initially formed, while the carbon emission efficiency of most cities in Anhui Province is low, and there is no significant change compared with 2001. The reason for this is that Anhui Province was in a period of rapid industrialization and the consumption of fossil energy increased sharply, while the corresponding environmental protection facilities were slow to be upgraded and the economic development was sloppy, so the carbon emission efficiency was maintained at a low level. In contrast, during the same period, developed cities in southern Jiangsu and northern Zhejiang implemented industrial restructuring and development mode changes, and had more advanced environmental protection technology and equipment, a high efficiency of energy use, and significant economic benefits. Taking Yangzhou as an example, as a national historical and cultural city and an important tourist city, Yangzhou has gradually supported tourism as a regional pillar industry since 2007, and the tertiary industry with tourism as its core has been developed rapidly, with a significant change in industrial structure and a significant improvement in the city's carbon emission efficiency. In 2019, the spatial divergence of carbon emission

efficiency in the Yangtze River Delta urban agglomeration became more obvious, forming a structural feature with southern Jiangsu as the core decayed outward. The spatial and temporal evolution of the carbon emission efficiency at the city scale indicated that the "core−edge" structure of carbon emission efficiency in the Yangtze River Delta region was gradually forming, and the low-carbon development and green transformation of economically developed regions were increasingly prominent for the achievement of the overall carbon emission reduction target of the region and the leading role in the transformation of the development mode of the surrounding areas.

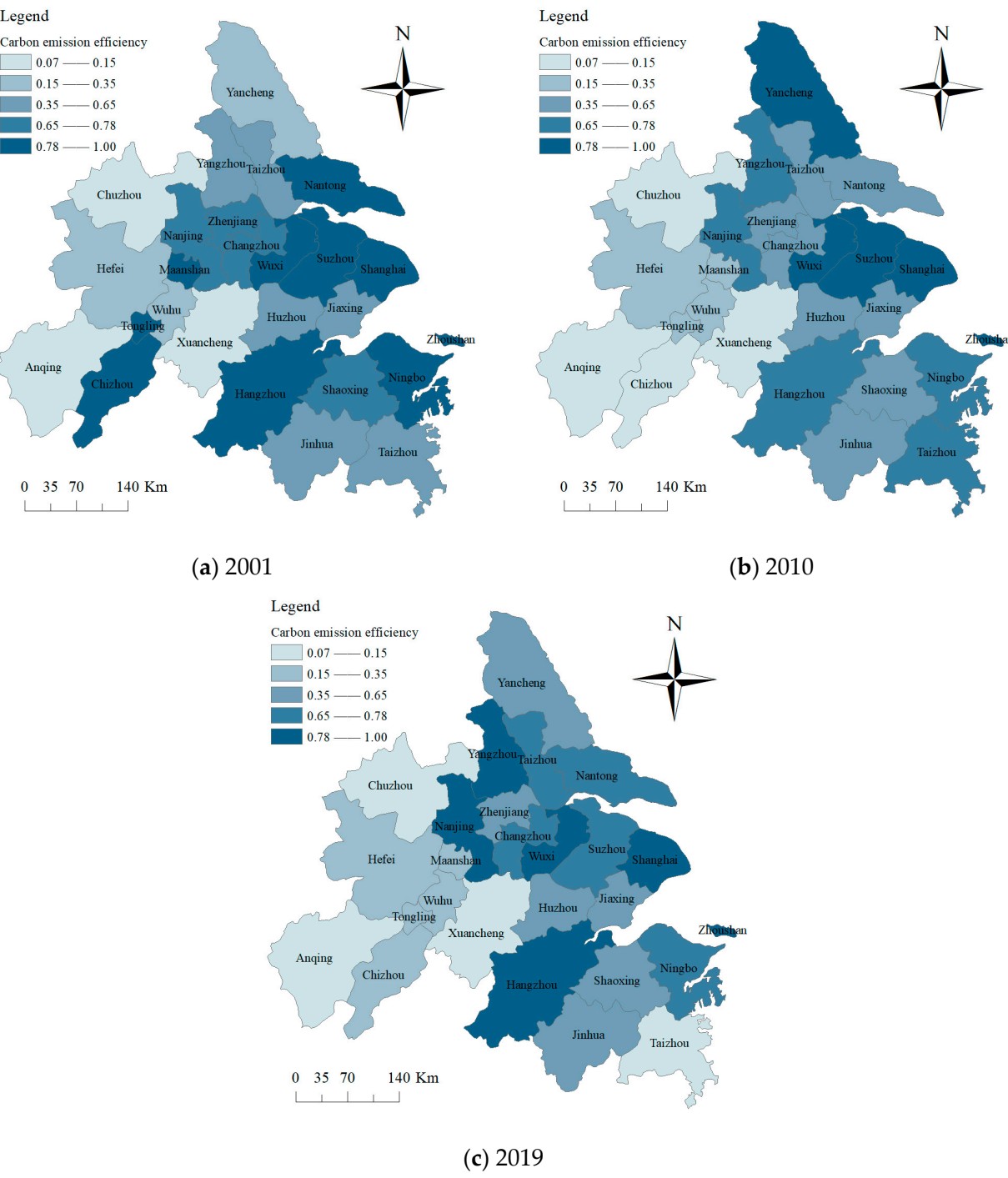

**Figure 4.** Spatial pattern of carbon emission efficiency in the Yangtze River Delta urban agglomeration from 2001 to 2019. (source: mapping through ArcGIS 10.8 software).

*5.2. Analysis of the Spatial Correlation Network Structure of Carbon Emission Efficiency in the Yangtze River Delta Urban Agglomeration*

5.2.1. Overall Network Structure Features

Based on the OD attribute data obtained from the spatial correlation matrix transformation, the cross-section data of 2001 and 2019 were selected, and the spatial correlation network map of the carbon emission efficiency was drawn using the ArcGIS visualization tool (Figure 5). As can be seen from Figure 5, the spatial correlation of carbon emission efficiency in the Yangtze River Delta urban agglomeration presents a complex network structure form with multiple thread and flow directions. With the evolution of time, the network gradually shows the core−edge structure characteristics with multiple centers (Table 3).

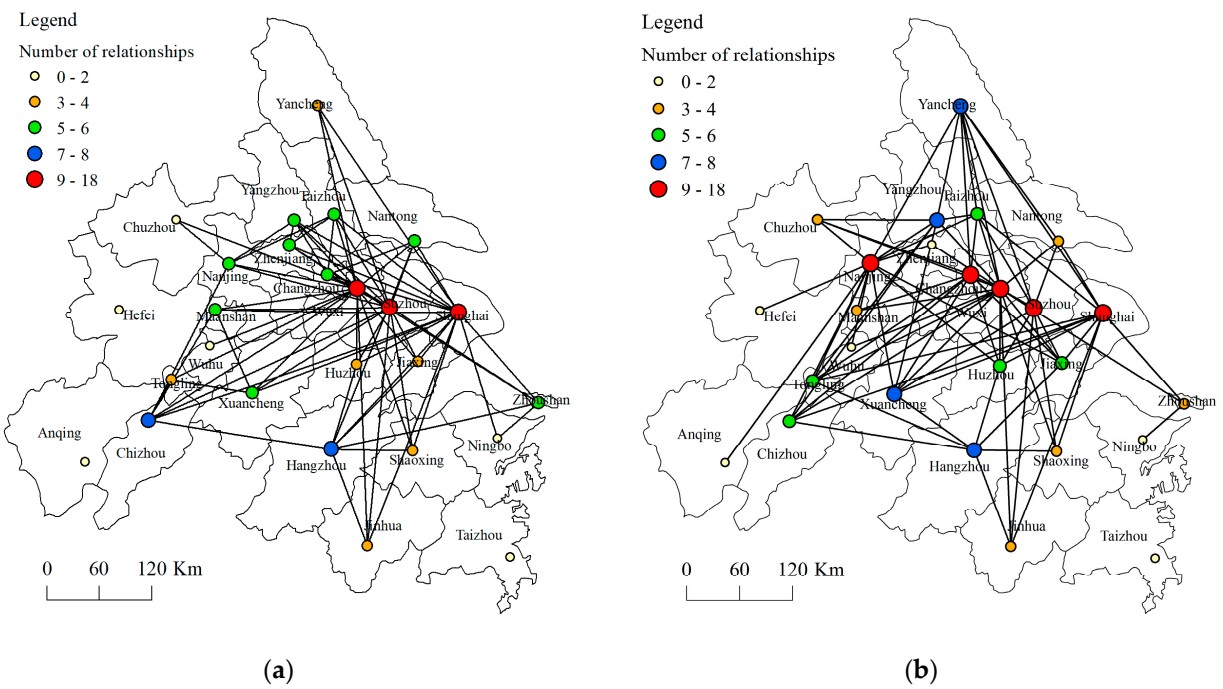

(**a**)                                                       (**b**)

**Figure 5.** Spatial correlation network of carbon emission efficiency in the Yangtze River Delta urban agglomeration. (**a**) 2001; (**b**) 2019.

**Table 3.** Composition of the core−edge structure of the carbon emission efficiency spatial network in the Yangtze River Delta urban agglomeration.

| Position | 2001 | 2019 |
|---|---|---|
| Core (number of associations ≥ 10) | Wuxi, Shanghai, and Suzhou | Shanghai, Wuxi, Nanjing, and Changzhou |
| Secondary core (number of associations ∈ [5, 10)) | Hangzhou, Chizhou, Nanjing, Changzhou, Yangzhou, Taizhou, Nantong, Zhenjiang, Zhoushan, Ma'Anshan, and Xuancheng | Suzhou, Yangzhou, Yancheng, Hangzhou, Xuancheng, Taizhou, Jiaxing, Huzhou, Tongling, and Chizhou |
| Secondary edge (number of associations ∈ [3, 5)) | Jiaxing, Huzhou, Shaoxing, Jinhua, Yancheng, and Tongling | Nantong, Shaoxing, Jinhua, Chuzhou, Zhoushan, and Ma'Anshan |
| Edge (number of associations ≤ 2) | Ningbo, Chuzhou, Wuhu, Taizhou, Hefei, and Anqing | Zhenjiang, Hefei, Anqing, Taizhou, Wuhu, and Ningbo |

In 2001, Wuxi, Shanghai, and Suzhou had the highest number of affiliations and were in the core position; Hangzhou, Chizhou, and Nanjing also had more affiliations and were in the secondary core position; and Wuhu, Taizhou, and Hefei had fewer affiliations

and were in the marginal position. In 2019, the complexity of the network increased and, in addition to Wuxi and Suzhou, Nanjing and Changzhou also entered the core circle, along with Yancheng and Xuancheng. In 2017, the complexity of the network increased and, except for Wuxi and Suzhou, Nanjing and Changzhou also entered the core circle; Yancheng, Xuancheng, and Yangzhou occupied the secondary core position; and Zhenjiang and Chizhou decreased and changed from the secondary core position to the edge position. Hefei, Anqing, Taizhou, and Wuhu remained at a low level, mainly because these cities are geographically at the edge of the Yangtze River Delta urban agglomeration and the neighboring cities are mostly low-ranking cities in the network, which have a low linkage effect with the surrounding areas. The linkage effect of these cities is low and the driving effect is not obvious. In general, the number of core and sub-core cities is relatively stable, with small changes during the study period, while the number of fringe and sub-fringe cities varies greatly, forming a core–fringe structure with southern Jiangsu as the core area, northern Zhejiang and central Jiangsu as the sub-core area, and central Anhui and southern Zhejiang as the fringe area (Figure 6).

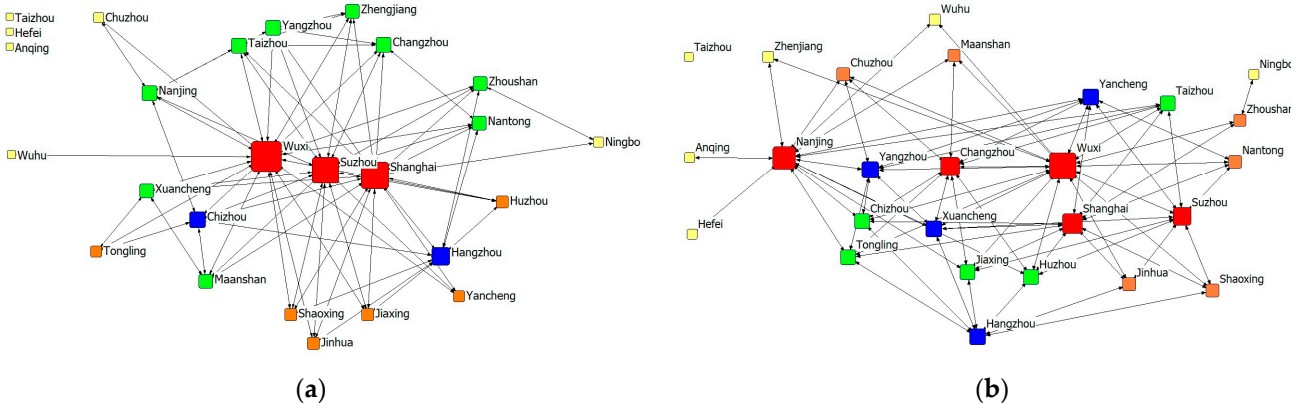

(**a**)  (**b**)

**Figure 6.** The core−edge structure pattern diagram of carbon emission efficiency in the Yangtze River Delta urban agglomerations. (**a**) 2001; (**b**) 2019.

With the help of Ucinet 6.0 software, the overall network structure characteristics of the carbon emission efficiency in the Yangtze River Delta urban agglomeration from 2001 to 2019 were calculated (Figure 7). The following can be seen: (1) The spatial network density value of carbon emission efficiency in the Yangtze River Delta urban agglomeration shows the characteristics of fluctuation and rise, rising from 0.215 to 0.248. The spatial correlation of the carbon emission efficiency is enhanced, but the close degree of correlation is still at a low level, so it is necessary to strengthen the exchange and cooperation and cross-regional coordination of the carbon emission reduction. (2) During the research period, the network correlation increased from 0.779 to 0.967, the correlation between cities increased, the number of "island" cities decreased, the accessibility between the network nodes was good, and the carbon emission efficiency had a significant spatial correlation and spillover effect among cities. (3) During the research period, the overall network efficiency was stable, with a small change range, and it was maintained at a high level, indicating that the carbon emission efficiency of the Yangtze River Delta urban agglomeration was less, the flow channel of the network nodes was more spacious, and the solidified structure had not yet been formed.

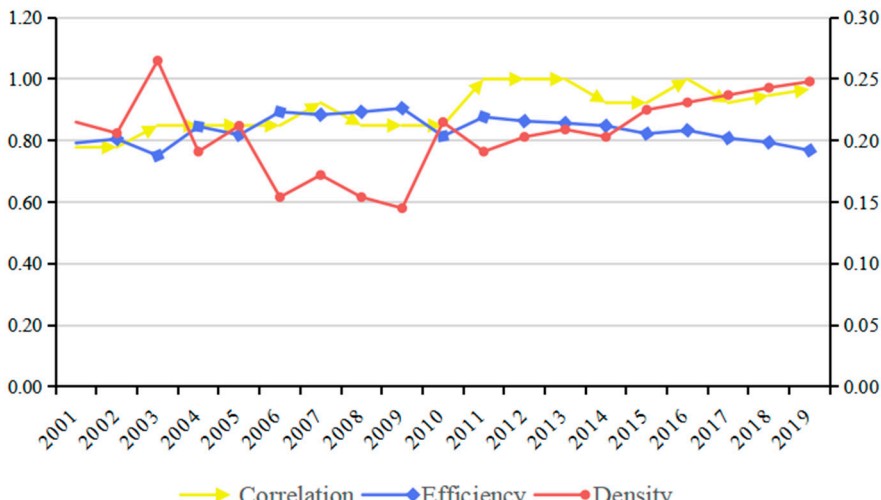

**Figure 7.** Overall network structure characteristic index of the carbon emission efficiency of the Yangtze River Delta urban agglomeration (source: calculated by using Ucinet 6.0 software).

5.2.2. Node Network Structure Features

The degree centrality, proximity centrality, and intermediary centrality were calculated using Ucinet 6.0 software (Appendix B).

(1) The degree centrality. The average values of the number centrality in 2001 and 2019 were 5.385 and 5.769, respectively. The cities with larger values were Wuxi, Shanghai, Suzhou, Nanjing, and Changzhou, which had more correlations with other cities in the correlation network and occupied a dominant position in the correlation network. The cities with lower values were Taizhou, Hefei, and Anqing, which were in a subordinate position in the correlation network. Wuxi, Shanghai, Suzhou, and Nanjing, as well as other cities, were the main spillovers of carbon emission efficiency, which had a "siphon effect" on other cities [46].

(2) The mean value of proximity centrality in 2001 was 56.385, which was 57 in 2019, with a small overall change. The reason for the higher proximity to the center of Taizhou, Hefei, and Anqing was that these cities were in the peripheral position in the carbon emission efficiency circle structure and the spatial structure of the Yangtze River Delta urban agglomeration; because of the influence of spatial distance and economic development, the carbon emission efficiency of these cities was controlled by other cities to a small extent, and was only correlated with neighboring cities in the periphery. Furthermore, the correlation with most nodes in the network was low, so it was necessary to improve the synergistic participation in emission reduction.

(3) The intermediary centrality. The average value of intermediary centrality degree in 2001 and 2019 was 7.385 and 11.192, respectively. The leading role of the network center nodes was significantly enhanced, and the unbalanced features of the network structure were further revealed. The intermediary center of Wuxi, Shanghai, Suzhou, and Nanjing was much greater than that of other cities. These cities had strong control over the spatial correlation of the carbon emission efficiency in other cities, and had obvious "intermediary" and "bridge" roles in the spatial correlation network. The intermediary center of Taizhou, Hefei, Anqing, and other cities was maintained at a low value level, with weak control over the spatial correlation of other cities, and it was easy to change their spatial correlation in cities with high intermediary centrality. In general, the distribution of intermediary centrality among cities was extremely unbalanced, and the polarization characteristics were significant.

### 5.3. Spatial Clustering Features

We analyzed the group network characteristics of carbon emission efficiency in the Yangtze River Delta urban agglomeration using a block model. The spatial correlation of each plate was as follows (Table 4).

**Table 4.** Division of carbon emission efficiency space plate in the Yangtze River Delta urban agglomeration (source: calculated from block model).

| Plate Type | City | Total Number of Accepted Relationships | | Total Number of Overflow Relationships | | Expected Internal Relationship Ratio | Actual Internal Relationship Ratio |
|---|---|---|---|---|---|---|---|
| | | Within the Plate | Outside the Plate | Within the Plate | Outside the Plate | | |
| Plate I | Shanghai, Wuxi, Suzhou, and Nanjing | 5 | 84 | 3 | 20 | 16.26 | 13.04 |
| Plate II | Changzhou, Yancheng, Hangzhou, Yangzhou, Xuancheng, Huzhou, Taizhou, and Shaoxing | 21 | 26 | 27 | 36 | 48.90 | 42.86 |
| Plate III | Nantong, Jiaxing, Jinhua, Zhoushan, Tongling, Maanshan, and Ningbo | 16 | 17 | 11 | 55 | 34.17 | 16.67 |
| Plate IV | Zhenjiang, Hefei, Anqing, Taizhou, Chizhou, Wuhu, and Chuzhou | 3 | 18 | 4 | 34 | 12.14 | 10.53 |

According to the results, Shanghai, Suzhou, Wuxi, and Nanjing belonged to plate I; Changzhou, Yancheng, Hangzhou, Yangzhou, Xuancheng, Huzhou, Taizhou, and Shaoxing belonged to plate II; Nantong, Jiaxing, Jinhua, Zhoushan, Tongling, Maanshan, and Ningbo belonged to plate III; and Zhenjiang, Hefei, Anqing, Taizhou, Chizhou, Wuhu, and Chuzhou belonged to plate IV.

Plate I had 89 spillover relationships, of which 84 were external to the plate; the proportion of actual internal relationships was smaller than the proportion of desired internal relationships; and the number of relationships received by the plate from other plates was much greater than the number of relationships that spilled outwards, indicating that plate I was a "net beneficiary plate". Plate II had 63 spillover relationships, 27 of which were internal to the plate, and the proportion of expected internal relationships (48.90%) was greater than the proportion of actual internal relationships (42.86%), which means that this segment had spillover effects both internally and externally, and was a "two-way spillover plate". Plate III had 66 spillover relationships and received 11 spillover relationships from other plates, and the proportion of expected internal relationships (34.17%) was significant. The proportion of desired internal relationships (34.17%) was significantly greater than the proportion of actual internal relationships (16.67%), and the number of overflowing relationships was significantly greater than the number of receiving relationships, so it was a "net overflow plate". Plate IV had 38 overflowing relationships and 21 receiving relationships, and the proportion of desired internal relationships (12.14%) was greater than the proportion of actual internal relationships (10.53%), which means that this segment was both outward spilling and receiving spillover from other plates, and was thus a "broker plate".

Based on the above results, we created the plate relationship diagram (Figure 8).

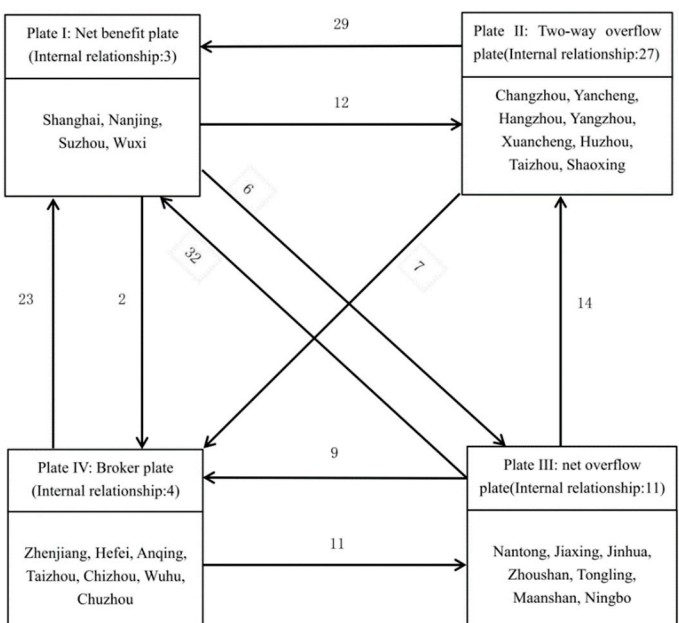

**Figure 8.** Correlation diagram of the carbon emission efficiency plate in the Yangtze River Delta urban agglomeration (source: calculated from the block model).

In general, Shanghai, Wuxi, Suzhou, and Nanjing, as well as other cities, have received large overflow relationships from other plates in the carbon emission efficiency correlation network, and are the beneficiaries of the network. The reason for this is that the huge economic size of these cities is highly attractive to low-carbon production factors such as capital, technology, and talent [47], which have a significant polarization effect on the surrounding cities. The number of spillover relationships in Nantong, Jiaxing, Jinhua, Tongling, Maanshan, and Ningbo is greater than the number of receiving relationships, showing distinct net spillover characteristics, which is the loss-making party in the network. Zhenjiang, Wuhu, Chuzhou, and other cities, through receiving and spillover relationships, strengthen the connection between the net overflow sectors and the net benefit sectors, thus becoming the intermediary of the net spillover and net benefit sector. Changzhou, Yancheng, Hangzhou, Yangzhou, Xuancheng, Huzhou, Taizhou, Shaoxing, and other cities have produced spillover effects to both the net benefit plate and the broker plate and receive the net benefit plate's overflow; at the same time, the contacts between the internal members are also frequent, regulating the connection between the plates to some extent, which is the regulator in the network.

### 5.4. Analysis of Influencing Factors

#### 5.4.1. QAP Correlation Analysis

The QAP method based on the secondary assignment procedure was used to calculate the correlation coefficient between each of the influencing factors and the spatial correlation structure of carbon emission efficiency in the Yangtze River Delta urban agglomeration (Table 5). From the table, the energy consumption per unit GDP, government environmental regulation, R&D expenditure, and the import and export trade correlation coefficient passed the significance level test of 1%, indicating that the energy structure, environmental regulation, technical level, and economic extroversion were the main factors of the Yangtze River Delta urban agglomeration's carbon emission efficiency's space correlation network structure. The correlation coefficients of the energy structure, industrial structure, and per capita green space area were negative, but not significant, indicating that they had no obvious effect on the formation of the spatial correlation network of the carbon emission efficiency.

**Table 5.** QAP correlation analysis and regression analysis results.

| Variable | Correlation Analysis | | Regression Analysis | |
|---|---|---|---|---|
| | Coefficient | *p*-Value | Coefficient | *p*-Value |
| Differences in energy resource structure | −0.037 | 0.202 | −0.003 | 0.158 |
| Differences in environmental regulation | 0.378 *** | 0.003 | 0.282 ** | 0.004 |
| Differences in energy consumption per unit of GDP | 0.292 ** | 0.038 | 0.224 ** | 0.035 |
| Differences in secondary industry | −0.072 | 0.202 | −0.004 | 0.160 |
| Differences in R&D spending | 0.392 ** | 0.002 | 0.300 ** | 0.003 |
| Differences in per-capita green space area | −0.004 | 0.328 | 0.012 | 0.153 |
| Differences in total import and export trade | 0.353 ** | 0.023 | 0.259 ** | 0.033 |

Note: ** $p < 0.05$, *** $p < 0.01$; $R^2 = 0.583$. The level of significance is 0.000.

### 5.4.2. QAP Regression Analysis

In order to avoid the bias in the regression results caused by the multicollinearity among independent variables, a QAP regression analysis was conducted using Ucinet 6.0 software to set the number of random permutations to 10,000, and the regression results are shown in Table 5. The regression equation had an overall fit level of 0.583, indicating that seven types of factors, including differences in the energy structure, differences in environmental regulations, differences in energy consumption per unit of GDP, differences in the proportion of secondary industries, differences in R&D expenditure, differences in per-capita green space, and differences in total import and export trade, could explain 58.3% of the spatial correlation of the carbon emission efficiency in the Yangtze River Delta urban agglomeration well.

Specifically, the regression coefficient of the environmental regulation difference (R = 0.282, $p < 0.01$) was significantly positive at a 1% level, indicating that the difference in the strength of government environmental regulation promoted the formation of the spatial correlation of the carbon emission efficiency in the Yangtze River Delta urban agglomeration; this is mainly because cities with relatively strict environmental supervision pressure tend to conduct a certain degree of "carbon transfer" to cities with less environmental control [45]. In reality, there are great differences in carbon emission control in different cities in the Yangtze River Delta urban agglomeration, which objectively promote the formation of the spatial correlation network of carbon emission efficiency. The difference in energy consumption per unit of GDP (R = 0.224, $p < 0.05$) suggests that, the greater the difference in the combined energy use efficiency and technology level, the more it helps to establish low-carbon development linkages between cities. The efficiency of comprehensive energy use is closely related to the local technological level, while resource development, production exchange, and the mobility of technicians are more frequent between regions with a greater difference in technological level, facilitating the formation of spatial linkages. The regression coefficient of the R&D investment's difference (R = 0.300, $p < 0.01$) is significantly positive at the 5% level, which means that the cities with a greater difference in technology research are more likely to have spatial correlation with carbon emission efficiency. R&D investment is directly related to the level of urban economic development. The widening differences in low-carbon energy efficiency technologies between cities have exacerbated the disparities in the level of low-carbon resource development and human capital, which can easily lead to the spillover and absorption of low-carbon resources across regions. In addition, in order to coordinate regional development, in recent years, the government has encouraged the exchange of scientific and technical personnel and technical services between cities with widely differing levels of technology, which has further enhanced carbon efficiency linkages. The regression coefficient of the total import and export trade's difference (R = 0.259, $p < 0.05$) is significantly positive at the 5% level, which implies that a relatively close level of trade is not conducive to a correlation

between cities in terms of carbon emission efficiency. The reason for this is that a relatively close level of trade implies that these cities are at a similar stage of development, have a similar demand for the resource elements needed to improve carbon efficiency, and have a competitive relationship for the production of trade goods.

The regression coefficients of the differences in energy structure, industrial structure, and per-capita green space are negative, but not significant, indicating that their inter-city differences cannot yet significantly influence the formation of a spatial correlation network of carbon emission efficiency in the Yangtze River Delta urban agglomeration. This may be because the current industrial and energy consumption structures in the Yangtze River Delta region are relatively stable, and the inter-city differences in the energy structure and industrial structure are not significant, which has a weak degree of explanation regarding the impact on carbon emission efficiency; at the same time, the per-capita green area mainly reflects people's living standards and urban greening, and the urban greening industry has a similar degree of development, which in turn weakens its influence on the spatial correlation relationship of the carbon emission efficiency in the Yangtze River Delta region.

## 6. Conclusions and Policy Implications

This paper measures the carbon emission efficiency of the Yangtze River Delta urban agglomeration from 2001 to 2019 using the U-S SBM model, portrays the spatially linked network structure of the carbon emission efficiency of the Yangtze River Delta urban agglomeration with the help of social network analysis, analyzes the spatio-temporal evolution characteristics, discusses the influencing factors using the QAP regression method, and obtains the following conclusions:

(1)     During the study period, the carbon emission efficiency of the Yangtze River Delta urban agglomeration as a whole showed a fluctuating downward trend, with the average value changing from 0.671 in 2001 to 0.522 in 2019. The spatial distribution gradually stabilized and showed significant spatial divergence, with the overall pattern being high in the east and low in the west. In 2019, the spatial divergence of the carbon emission efficiency of the Yangtze River Delta urban agglomeration became more obvious, and gradually formed a structural feature with southern Jiangsu as the core and declining outward, with the structural characteristics of the core decaying outward.

(2)     During the research period, the spatial correlation fluctuation of the carbon emission efficiency in the Yangtze River Delta urban agglomeration was enhanced, showing a complex network structure with a multi-threaded and multi-flow direction. With the evolution of time, the southern Jiangsu region as the core area, northern Zhejiang and central Jiangsu region as the sub-core area, and central Anhui and southern Zhejiang region as the edge of the core–marginal structure were gradually formed.

(3)     In the spatial correlation network of the carbon emission efficiency of the Yangtze River Delta urban agglomeration, the degree centrality and intermediary centrality of Shanghai, Wuxi, Nanjing, and Suzhou were much higher than those of the other 22 cities, and had a strong influence and control over the spatial correlation of the carbon emission efficiency of other cities, playing the role of the bridge and hub of network connection. In 2019, the spatial correlation of Changzhou was significantly higher, and its control power gradually emerged. Anqing, Taizhou, and other cities had a weaker correlation with the carbon emission efficiency of most cities in the network and became "islands".

(4)     The 26 cities in the Yangtze River Delta urban agglomeration can be divided into four plates, "net benefit", "net overflow", "two-way overflow", and "broker", which have a strong spatial linkage effect between each plate. Shanghai, Wuxi, Nanjing, and Suzhou belong to the net beneficiary sectors, and produced a significant siphon effect on the low-carbon production factors in the surrounding cities. Changzhou, Yancheng, and other cities not only produced spillover effects to the net benefit plate

and the broker plate, but also receive the spillover of the net benefit plate, which were the regulators in the network.

(5) Government environmental regulation, energy efficiency, technology research and development, and economic divergence in the Yangtze River Delta urban agglomeration carbon efficiency space significantly influenced the formation of the correlation network. The energy consumption per unit GDP, industrial structure, and per-capita green space area differences on the Yangtze River Delta urban agglomeration carbon emission efficiency space correlation network structure were not significant.

According to the research results, the following policy implications can be obtained:

(1) The carbon emission efficiency of the Yangtze River Delta urban agglomeration is spatially linked, and each city's carbon emission efficiency is not only dependent on its own development, but also influenced by other cities in its vicinity. At present, the degree of carbon emission efficiency linkage in the Yangtze River Delta urban agglomeration is relatively loose, with large disparities within the region. Therefore, local governments must deepen reforms, establish cooperation and exchange mechanisms in various areas [48], strengthen the linkage of energy conservation and emission reduction in the region through macro-control means, and continuously adjust and optimize the spatial network structure of carbon emission efficiency.

(2) Each city should adopt differentiated measures according to its position in the spatial network of carbon efficiency to promote the improvement of the overall carbon efficiency of the Yangtze River Delta urban agglomeration. Shanghai, Wuxi, Suzhou, and Nanjing have a high carbon efficiency and are located at the core, but they are "net beneficiaries" and have a siphoning effect on neighboring cities. These cities should take on the responsibility of leading the region in reducing emissions and should play a leading role in providing financial, technical, and human resources support to other cities [49]. Taizhou, Hefei, Anqing, and other cities are located at the edge of the network and face the loss of carbon emission efficiency caused by the outflow of resource factors. These cities should seize the opportunity of the integration of the Yangtze River Delta; actively integrate into the carbon emission efficiency correlation network of the Yangtze River Delta urban agglomeration; introduce capital, talents, technology, and other resource factors from cities with high carbon emission efficiency; and improve their own carbon emission efficiency.

(3) The formation of the spatial association of the carbon emission efficiency in the Yangtze River Delta urban agglomeration is influenced by a number of factors. These driving factors should be fully utilized to enhance the spatial association of carbon emission efficiency. On the one hand, cities should enhance the support of information and transportation networks for the circulation of carbon emission efficiency support factors, and pay attention to the optimization of energy structures and the improvement of technology levels [50]; on the other hand, they should strengthen intra-regional trade ties, bring into play the advantages of cities with a higher carbon emission efficiency in energy use and technology research, and strengthen the export of green products and technologies, so as to promote the sustainability of the region through innovation.

Compared with existing studies on regional carbon emission efficiency, this paper expands the research perspective and methodology, and provides spatial insights into the solution of regional carbon efficiency problems. However, because of the limitations of data acquisition and other factors, there are still some limitations in this paper. Firstly, the factors affecting the spatial correlation of regional carbon emission efficiency are too complex and difficult to fully address in this paper; in addition, the impact of the COVID-19 pandemic on the correlation of carbon emission efficiency is not explored in depth in this paper, and the impact of the COVID-19 pandemic on regional linkages and regional sustainability needs to be explored in subsequent studies.

**Author Contributions:** Conceptualization, C.L.; methodology, C.L.; software, R.T. and Y.G.; formal analysis, C.L. and R.T.; investigation, C.L. and Y.S.; writing—original draft preparation, R.T. and Y.G.; writing—review and editing, C.L. and X.L.; supervision, C.L. All authors have read and agreed to the published version of the manuscript.

**Funding:** This research received no external funding.

**Institutional Review Board Statement:** Not applicable.

**Informed Consent Statement:** Not applicable.

**Data Availability Statement:** The basic data used in this research can be found on the websites of the cities' Bureau of Statistics, Statistical Yearbooks, NGDC, The National Ministry of Natural Resources, and other databases.

**Acknowledgments:** We sincerely thank the academic editors and anonymous reviewers for their kind suggestions and valuable comments.

**Conflicts of Interest:** The authors declare no conflict of interest.

## Appendix A

**Table A1.** Carbon emission efficiency in the Yangtze River Delta urban agglomeration from 2001 to 2019.

| City | 2001 | 2003 | 2005 | 2007 | 2009 | 2011 | 2013 | 2015 | 2017 | 2019 |
|------|------|------|------|------|------|------|------|------|------|------|
| Shanghai | 1.00 | 1.00 | 1.00 | 1.00 | 1.00 | 1.00 | 1.00 | 1.00 | 1.00 | 1.000 |
| Nanjing | 0.779 | 1.000 | 1.000 | 0.851 | 0.858 | 1.000 | 1.000 | 1.000 | 0.995 | 1.000 |
| Wuxi | 1.000 | 1.000 | 1.000 | 1.000 | 1.000 | 1.000 | 1.000 | 0.855 | 0.859 | 1.000 |
| Changzhou | 0.757 | 0.735 | 0.660 | 0.677 | 0.734 | 0.736 | 0.761 | 0.717 | 0.735 | 0.723 |
| Suzhou | 0.884 | 1.000 | 1.000 | 0.796 | 1.000 | 0.717 | 0.785 | 0.702 | 0.643 | 0.635 |
| Nantong | 1.000 | 0.762 | 0.683 | 0.592 | 0.640 | 0.714 | 0.704 | 0.700 | 0.710 | 0.711 |
| Yancheng | 0.347 | 1.000 | 1.000 | 1.000 | 1.000 | 0.586 | 0.491 | 0.472 | 0.465 | 0.431 |
| Yangzhou | 0.640 | 0.619 | 0.705 | 0.732 | 1.000 | 1.000 | 1.000 | 1.000 | 1.000 | 1.000 |
| Zhenjiang | 0.727 | 0.711 | 0.723 | 0.696 | 0.754 | 0.699 | 0.649 | 0.587 | 0.543 | 0.503 |
| Taizhou | 0.486 | 0.513 | 0.561 | 0.573 | 0.635 | 0.647 | 0.644 | 0.615 | 0.631 | 0.586 |
| Hangzhou | 1.000 | 1.000 | 1.000 | 1.000 | 1.000 | 1.000 | 1.000 | 1.000 | 1.000 | 1.000 |
| Ningbo | 1.000 | 0.803 | 0.745 | 0.693 | 0.749 | 0.775 | 0.795 | 0.713 | 0.703 | 0.698 |
| Jiaxing | 0.633 | 0.665 | 0.589 | 0.475 | 0.448 | 0.468 | 0.446 | 0.379 | 0.347 | 0.372 |
| Huzhou | 0.576 | 0.521 | 0.494 | 0.448 | 0.504 | 0.530 | 0.520 | 0.468 | 0.410 | 0.398 |
| Shaoxing | 0.720 | 0.676 | 0.659 | 0.617 | 0.601 | 0.645 | 0.659 | 0.594 | 0.530 | 0.536 |
| Jinhua | 0.634 | 0.552 | 0.543 | 0.491 | 0.468 | 0.474 | 0.487 | 0.444 | 0.366 | 0.345 |
| Zhoushan | 1.000 | 1.000 | 1.000 | 1.000 | 1.000 | 1.000 | 1.000 | 1.000 | 1.000 | 1.000 |
| Taizhou | 0.458 | 0.432 | 0.336 | 0.318 | 0.292 | 0.302 | 0.248 | 0.164 | 0.149 | 0.138 |
| Hefei | 0.211 | 0.217 | 0.215 | 0.228 | 0.278 | 0.200 | 0.211 | 0.235 | 0.231 | 0.303 |
| Wuhu | 0.263 | 0.280 | 0.249 | 0.269 | 0.335 | 0.235 | 0.231 | 0.198 | 0.176 | 0.174 |
| Ma'Anshan | 1.000 | 0.465 | 0.997 | 1.000 | 1.000 | 0.281 | 0.249 | 0.218 | 0.210 | 0.193 |
| Tongling | 1.000 | 1.000 | 1.000 | 1.000 | 1.000 | 1.000 | 1.000 | 0.262 | 0.251 | 0.219 |
| Anqing | 0.077 | 0.072 | 0.067 | 0.066 | 0.079 | 0.080 | 0.077 | 0.100 | 0.101 | 0.104 |
| Chuzhou | 0.112 | 0.109 | 0.097 | 0.098 | 0.110 | 0.114 | 0.118 | 0.116 | 0.112 | 0.100 |
| Chizhou | 0.999 | 1.000 | 0.172 | 0.159 | 0.179 | 0.200 | 0.214 | 0.223 | 0.211 | 0.224 |
| Xuancheng | 0.153 | 0.136 | 0.114 | 0.114 | 0.124 | 0.136 | 0.143 | 0.144 | 0.143 | 0.143 |
| Average | 0.671 | 0.664 | 0.639 | 0.611 | 0.646 | 0.598 | 0.594 | 0.535 | 0.520 | 0.521 |

## Appendix B

**Table A2.** Central index of the carbon emission efficiency network in the Yangtze River Delta urban agglomeration.

| City | 2001 | | | 2019 | | |
|---|---|---|---|---|---|---|
| | Degree Centrality | Proximity Centrality | Intermediary Centrality | Degree Centrality | Proximity Centrality | Intermediary Centrality |
| Shanghai | 16 | 42 | 46.250 | 11 | 49 | 26.781 |
| Nanjing | 6 | 51 | 3.601 | 14 | 45 | 68.464 |
| Wuxi | 18 | 38 | 79.882 | 18 | 38 | 89.495 |
| Changzhou | 6 | 50 | 1.073 | 10 | 47 | 9.853 |
| Suzhou | 15 | 42 | 30.517 | 9 | 48 | 8.493 |
| Nantong | 5 | 51 | 1.460 | 4 | 54 | 0.400 |
| Yancheng | 3 | 53 | 0.406 | 7 | 49 | 8.458 |
| Yangzhou | 6 | 50 | 1.699 | 8 | 47 | 4.327 |
| Zhenjiang | 5 | 51 | 0.573 | 2 | 55 | 0.773 |
| Taizhou | 6 | 50 | 1.699 | 6 | 50 | 4.458 |
| Hangzhou | 8 | 50 | 5.250 | 7 | 59 | 5.918 |
| Ningbo | 2 | 58 | 0.000 | 1 | 80 | 0.137 |
| Jiaxing | 4 | 52 | 0.960 | 6 | 49 | 6.740 |
| Huzhou | 4 | 52 | 0.960 | 6 | 49 | 6.740 |
| Shaoxing | 4 | 52 | 0.960 | 4 | 56 | 1.793 |
| Jinhua | 4 | 52 | 0.960 | 4 | 56 | 1.793 |
| Zhoushan | 5 | 51 | 6.193 | 3 | 57 | 23.275 |
| Taizhou | 0 | 100 | 0.000 | 0 | 150 | 0.000 |
| Hefei | 0 | 100 | 0.000 | 1 | 68 | 0.146 |
| Wuhu | 1 | 59 | 0.000 | 2 | 55 | 0.773 |
| Ma'Anshan | 5 | 51 | 0.606 | 3 | 54 | 1.106 |
| Tongling | 3 | 54 | 0.200 | 6 | 49 | 6.149 |
| Anqing | 0 | 100 | 0.000 | 1 | 68 | 0.125 |
| Chuzhou | 2 | 57 | 0.000 | 4 | 53 | 1.249 |
| Chizhou | 7 | 49 | 6.677 | 6 | 49 | 6.149 |
| Xuancheng | 5 | 51 | 2.073 | 7 | 48 | 7.812 |
| Average | 5 | 56 | 7.385 | 5.769 | 57 | 11.192 |

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
