# Peer review of "Research on the Structure of Carbon Emission Efficiency and Influencing Factors in the Yangtze River Delta Urban Agglomeration"

_sustainability, doi:10.3390/su14106114_

Round 1
Reviewer 1 Report
- I suggest the authors revise the introduction of the study per the comments raised. Authors can also use the following points below as a guideline to help them come out with an interesting introduction that is more scientific.
• Background & Significance: (What general background does the reader need in order to understand the manuscript and how important is it in the context of scientific research).
• Problem definition: (What are the research questions to fill in the gaps of the existing knowledge body or methodology).
• Motivations & Objectives: (Why are you conducting the study and what are goals to achieve?) - Conclusions of the study should be improved, where it is necessary to suggest the real policy recommendations and constructive solutions for the possible problems which were found.
- Policy implications are not strongly related to the research findings, so it needs to be improved.
- Include limitations and further research
Reviewer 2 Report
Review Report. For :
Article :
Research on structure of carbon emission efficiency and influencing factors in Yangtze River Delta urban agglomeration
In my opinion, you have to rewrite the keywords to remove the compound ones and reduce them, an entire phrase cannot be a keyword. The abstract is too long, it is 308 words, it must be reduced, it is not necessary to show all the conclusions in the abstract, showing the main conclusion in the abstract is enough. In the introduction, it is necessary to identify the main objective of the work, which is presumed to be to analyze CO2 emissions in the Yangtze River, as well as some secondary objective derived from it.
The research hypotheses have to be considered as such, h1, h2..........., the work is quite interesting, but without this the approach remains in the air.
Figure 1 does not include the source (in this case own elaboration), and it is not a figure, it is a graph. Nor do they put the source in figure 2 (it is understood that it is their own elaboration). In all the tables it is necessary to put the source. In the discussion of results, it is missing to compare the results with those of some study mentioned in the bibliography.
Otherwise the work is quite good, so congratulate the authors for its originality. Finally, it would be convenient to add more international jcr bibliography, practically all the bibliography is Asian.
Reviewer 3 Report
This work deals with a relatively current topic, namely the effectiveness of emission allowances, which are a basic tool for air protection.
In this work specifically in the Yangtze River Delta region, in China.
The authors chose statistical methods, namely the ultra-efficiency SBM model and the spatial correlation network analysis method.
These are known and frequently used statistical analyzes that are suitable for this type of research.
Time series must be set for this type of analysis. In this work, the authors calculate the time series from 2001 to 2017.
- This is where I see the biggest problem of this work, and that is the relative outdated data. In the current year 2022, data up to 2020 are already available statistically, or it is necessary to find them by searching at state administration bodies. In recent years, there has been a significant increase in the prices of emission allowances on world markets, and the price of emission allowances is a fundamental factor that affects issuers! Therefore, it can be stated that the results of this analysis are debatable in terms of topicality.
However, the work also shows other, less serious shortcomings.
- Tables, figures and graphs do not have the listed sources. Even in the case of own creation, it is necessary to state the source under the table - eg own work, calculation, etc.
- The text on lines 209 - 213 wraps around the parentheses and reduces the intelligibility of the statement.
- The statistical formulas used are not always sufficiently described. It is necessary that under each general formula there is a term Where: and all quantities and variables are described in detail so that they are understandable even to a layman.
- The authors should also reflect on the fact that in 2004-2008 the markets for emission allowances and emission production were affected by the global economic crisis and at least mention and quote this fact for the objectivity of work.
- The discussion lacks a specific comparison of results and a subsequent discussion with the work on the same topic, of which there is enough in the database, because it is a relatively frequent topic.
- At the end of the work it is necessary to state the specific results achieved in numbers, just a general comment is not enough.
Round 2
Reviewer 1 Report
The author(s) have incorporated the comments and should be accepted for publication.
Reviewer 3 Report
Authors resolved all stated notes and they had improved their work.
Author Response
Thank you very much for your valuable comments.